# Diagnosis of Malnutrition According to GLIM Criteria Predicts Complications and 6-Month Survival in Cancer Outpatients

**DOI:** 10.3390/biomedicines10092201

**Published:** 2022-09-06

**Authors:** Marta Gascón-Ruiz, Diego Casas-Deza, Maria Marti-Pi, Irene Torres-Ramón, María Zapata-García, Andrea Sesma, Julio Lambea, María Álvarez-Alejandro, Elisa Quilez, Dolores Isla, Jose Miguel Arbonés-Mainar

**Affiliations:** 1Medical Oncology Department, University Hospital Lozano Blesa, Av San Juan Bosco 15, 50009 Zaragoza, Spain; 2Instituto de Investigación Sanitaria (IIS) de Aragón, Paseo Isabel la Católica 1-3, 50009 Zaragoza, Spain; 3Gastroenterology and Hepatology Department, University Hospital Miguel Servet, Paseo Isabel la Católica 1-3, 50009 Zaragoza, Spain; 4Translational Research Unit, University Hospital Miguel Servet, Instituto Aragonés de Ciencias de la Salud (IACS), Paseo Isabel la Católica 1-3, 50009 Zaragoza, Spain; 5Biomedical Research Center in Physiopathology of Obesity and Nutrition (CIBERon), Health Institute Carlos III (ISCIII), Av. Monforte de Lemos, 3-5, 28029 Madrid, Spain

**Keywords:** cancer, survival, GLIM, malnutrition, pain

## Abstract

**Background and Aims**: Malnutrition is a condition that has a great impact on oncology patients. Poor nutritional status is often associated with increased morbidity and mortality, increased toxicity, and reduced tolerance to chemotherapy, among other complications. The recently developed GLIM criteria for malnutrition aim to homogenize its diagnosis, considering the baseline disease status. We aimed to evaluate the performance of these new criteria for the prediction of complications and mortality in patients with cancer. **Methods**: This work is a prospective, single-center study. All outpatients under active treatment for head and neck, upper gastrointestinal, and colorectal tumors between February and October 2020 were recruited. These patients were followed up for 6 months, assessing the occurrence of complications and survival based on GLIM diagnoses of malnutrition. **Results**: We enrolled 165 outpatients, 46.66% of whom were malnourished. During the 6-month follow-ups, patients with malnutrition (46.7%, according to GLIM criteria) had a ~3-fold increased risk of hospital admission (*p* < 0.001) and occurrence of severe infection (considered as those requiring hospitalization, intravenous antibiotics, and/or drainage by interventional procedures) (*p* = 0.002). Similarly, malnourished patients had a 3.5-fold increased risk of poor pain control and a 4.4-fold increased need for higher doses of opioids (both *p* < 0.001). They also had a 2.6-fold increased risk of toxicity (*p* = 0.044) and a 2.5-fold increased likelihood of needing a dose decrease or discontinuation of cancer treatment (*p* = 0.011). The 6-month survival of malnourished patients was significantly lower (*p* = 0.023) than in non-malnourished patients. **Conclusions**: Diagnoses of malnutrition according to the GLIM criteria in oncology patients undergoing active treatment predict increased complications and worse survival at 6-month follow-ups, making them a useful tool for assessing the nutritional status of oncology patients.

## 1. Introduction

Malnutrition is a highly prevalent pathology in the oncology population, with a major negative impact both in terms of the risk of complications and overall survival [1,2,3]. In cancer patients, tumor-induced malnutrition is called cachexia–anorexia syndrome. This phenomenon constitutes the body’s inflammatory response against the active tumor, releasing a cascade of cytokines. This induces increased energy consumption with secondary catabolism of adipose tissue and, ultimately, of muscle tissue [3,4,5,6].

Weight loss is the major cause of morbidity and mortality in cancer patients and cachexia is the cause of death in one in three cancer patients [7,8,9]. Other effects of malnutrition include delayed healing of surgical wounds, malabsorption due to the lack of synthesis of digestive enzymes, increased treatment-associated immunosuppression, increased infectious complications, impaired quality of life, and decreased tolerance to treatment with increased treatment-related toxicities [10,11,12].

In terms of the diagnostic criteria, until a few years ago, there were no unified criteria, and several options were available. With the aim of standardizing the diagnosis of malnutrition in routine clinical practice, several nutrition societies developed new diagnostic criteria in 2019, known as the GLIM (Global Leadership Initiative on Malnutrition) criteria [13]. This is a two-step approach in which the risk of malnutrition is first evaluated and then, those patients at risk that meet at least one phenotypic criterion (weight loss, low body mass index, or low fat-free mass index), as well as one etiological criterion (decreased intake, malabsorption, or inflammatory condition), are diagnosed as malnourished.

In recent years, several studies have assessed the diagnostic ability of the criteria in oncology patients. These studies have favored the use of the GLIM criteria given their effectiveness in the early diagnosis of malnutrition [14,15,16,17,18]. However, only a few studies to date have assessed the ability of the new GLIM criteria to predict mortality in cancer patients. Furthermore, most of these studies are retrospective [16,19,20,21,22] or include very specific populations, such as hospitalized patients [23,24], or specific tumor sites, such as esophagogastric or lung tumors [17,20,22].

Therefore, the aim of our study was to globally and prospectively evaluate the ability of the GLIM criteria to predict mortality in oncological outpatients. For this purpose, three tumor sites were selected according to two different malnutrition risk profiles: head and neck and upper digestive tract tumors (esophagus, gastric, and pancreas) were selected as cancers with a high incidence of malnutrition. Colorectal cancer was chosen as a tumor with a lower incidence of malnutrition.

## 2. Materials and Methods

### 2.1. Ethical Considerations

The clinical research ethics committee of Aragon (CEICA) evaluated and approved the project with study code PI19/494. Informed consent was collected from all patients who agreed to participate in the study.

### 2.2. Design and Selection Criteria

An observational, prospective, single-center, non-interventional study was carried out in the Medical Oncology department of University Hospital Lozano Blesa, a tertiary level hospital in Zaragoza (Spain). Between February and October 2020, we consecutively recruited all patients undergoing follow-ups in the medical oncology outpatient clinics who underwent active treatment for a tumor in one of the following locations: head and neck, upper digestive tract, and/or colorectal.

The enrolled patients were followed prospectively from their inclusion in the study for the next six months or until death, whichever came first. Follow-up was adapted to the follow-up intervals in accordance with standard clinical practice, ranging from one to three weeks, depending on the different treatments.

Exclusion criteria were pregnancy, the coexistence of a neuropsychiatric condition that significantly interfered with the correct completion of the questionnaires, refusal by the patient to sign the informed consent form, and age under 18 years.

### 2.3. Patient Characteristics and Malnutrition Criteria

At the first visit (baseline), both demographic and tumor-related patient characteristics were collected: age, sex, tumor location, stage (localized vs metastatic), and current disease status (following the RECIST v1.1 solid lesion response evaluation criteria) [25] according to the latest available imaging tests (response/stability/progression), as well as current oncologic treatments (chemotherapy, radiotherapy, immunotherapy, targeted therapy, or combination) and current treatment times.

Nutritional assessment and diagnosis of malnutrition were performed using the GLIM criteria as a reference. These criteria were proposed in 2019 by the main scientific nutrition societies and require establishing a diagnosis of malnutrition with at least one phenotypic criterion (weight loss, low body mass index (BMI), and/or low free body mass index (FFMI)) and one etiological criterion (decreased intake, a chronic gastrointestinal condition, or systemic inflammation). Differentiation between moderate and severe malnutrition was made according to different cut-off points suggested for each phenotypic criterion in the consensus document.

BMI was calculated after an in-office measurement of the patient’s height and mass. According to the GLIM criteria classifications of moderate and severe malnutrition, the cut-off points used were as follows: <18.5 kg/m^2^ for patients under 70 years of age and <20 kg/m^2^ for patients over 70 years of age for the diagnosis of severe malnutrition and <20 kg/m^2^ for patients under 70 years of age and <22 kg/m^2^ for those over 70 years of age for moderate malnutrition. A bioimpedance analysis (Tanita DC 580 segmental; Tanita, Tokyo, Japan) was used to assess muscle mass.

Regarding the etiological criteria, a decrease in intake was considered when the patient presented a reduction of at least 50% of the baseline amount for at least one week or a reduction in any proportion for at least two weeks. Dysphagia, mechanical obstruction, exocrine pancreatic insufficiency, short bowel syndrome, chronic diarrhea (defined by a duration of at least 4 weeks), and any other cause of intestinal malabsorption were considered chronic gastrointestinal conditions.

Regarding the inflammatory status and following the recommendations of the GLIM criteria, the oncologic disease itself was not considered a sufficient condition for fulfilling this etiologic criterion. We considered this criterion fulfilled only if the tumor was in progression. The presence of other conditions, such as active infections or inflammatory processes of other origins, was also deemed sufficient.

The following variables were prospectively monitored during follow-up: death, need for emergency care and/or hospitalization, severe infection (considered as those requiring hospitalization, intravenous antibiotics, and/or drainage by interventional procedure), mild infections, treatment-related toxicity, changes in the treatment dose, poor pain control, need to increase analgesic treatment, and tumor progression. For all these variables, the date of the first event was collected.

### 2.4. Statistical Analysis

Continuous variables were summarized as the median (interquartile range [IQR]) or mean (standard deviation) or the number of cases (percentages). Categorical data were summarized by showing the number of individuals who fell into each category (e.g., malnutrition) and creating a relative frequency. The relative frequency of a given category is the frequency (number of individuals in that category) divided by the total sample size, multiplied by 100 to obtain the percentage. Continuous variables were tested for Gaussian distribution by the Shapiro–Wilk normality test and comparisons of the variables were performed using student’s t- or Mann–Whitney tests for normal- and non-normal-distributed variables, respectively. The χ^2^ test with Yates’s correction for continuity was used for comparison of the categorical variables. An ordinal regression model was used to determine the association between the clinical outcomes and nutritional status. Multivariable analysis was performed by adjusting for age and tumor localization. The Kaplan–Meier curve was plotted to estimate the probability of survival during follow-up. Data were analyzed using R version 4.0.3 (http://www.r-project.org, accessed on 30 March 2022). *p* values < 0.05 were considered significant.

## 3. Results

### 3.1. Patient Characteristics and Malnutrition Prevalence

A total of 165 patients (35.2% women) were included with a median age (interquartile range) of 67. Colorectal tumors were the most prevalent (49.7%), followed by those in the upper gastrointestinal tract (38.8%) and the head and neck area (11.5%) (Table 1). Most of the participants were receiving chemotherapy alone (83%) or chemotherapy combined with radiotherapy (12.1%). Immunotherapy or targeted therapy represented less than 5% of the treatments. The median duration of the treatments was 3 months. Most patients (69.1%) were in a metastatic stage of their oncologic disease, with 27.9% in progression, 41% in stable disease, and 30.9% in response.

The average BMI of the participants was 26.1 kg/m^2^ and 17% presented some GI condition, pancreatic insufficiency being the most abundant (6.7%). According to the GLIM criteria, the prevalence of malnutrition in the entire cohort was 53.3%, being moderate in 23.6% and severe in 23%. However, malnutrition changed widely depending on the tumor location (Figure 1), ranging from 34% (16% moderate, 18% severe) in colorectal patients to 57% (36% moderate, 21% severe) in individuals with head and neck tumors and 59% (29.5% moderate, 29.5% severe) in those with upper GI cancer.

As the GLIM consensus is a two-step model, the diagnosis of malnutrition was conducted by assessing the phenotypic and etiologic criteria. To explain the wide differences observed in the prevalence of malnutrition, we investigated whether there were differences between those criteria depending on the tumor location.

The presence of at least one phenotypic criterion, either non-volitional weight loss, low BMI, or reduced muscle mass, was found in 68%, 47%, and 75% of individuals with tumors in the head and neck, colorectal, and upper GI areas, respectively (*p* = 0.01). Those differences accounted for a disparate increased presence of non-volitional weight loss in patients with tumors in the upper GI region (Table 2). The etiologic criteria, namely reduced food intake or assimilation and inflammation were observed in 58%, 51%, and 72% of individuals with tumors in the head and neck, colorectal, and upper GI areas, respectively (*p* = 0.04). This difference is due to a much higher proportion of patients with tumors of the upper gastrointestinal tract with tumor progression and/or chronic gastrointestinal conditions.

### 3.2. Complications Resulting from Malnutrition

Patients with malnutrition according to the GLIM criteria had a 3-fold increased risk of admission (*p* < 0.001) and occurrence of severe infection during follow-up (*p* = 0.002). Similarly, malnourished patients had a 3.5-fold increased risk of poor pain control and a 4.4-fold increased need for a higher dose of opioids (both *p* < 0.001).

In terms of complications related to cancer treatment, patients with malnutrition had a 2.6-fold increased risk of toxicity (*p* = 0.044) and a 2.5-fold increased likelihood of needing a dose reduction or discontinuation of cancer treatment (*p* = 0.011). Those increased risks were independent of the tumor location as we obtained similar results when the ORs were calculated for each group (Appendix A).

Finally, in terms of oncological disease progression, the risk of tumor progression was 1.9 times higher in malnourished patients compared to non-malnourished patients (*p* = 0.049). Regarding survival, at six months after recruitment, 87.4% of non-malnourished patients were still alive compared to 71.4% of malnourished patients (OR 0.37, *p* = 0.012). These results can be seen in Table 3.

A Kaplan–Meier analysis (Figure 2) shows that there was a significant difference in survival according to the nutritional status of the oncology patients according to the GLIM criteria. Univariate Cox models confirmed that severe malnutrition was significantly associated with a ~3-fold increased risk of mortality in malnourished patients (HR = 2.98, 95%CI = 1.31–6.75, *p* = 0.003) compared to non-malnourished participants (HR = 1). Moderate malnutrition had a neutral effect on all-cause mortality in oncology patients (HR = 2.04, 95%CI = 0.85–4.92, *p* = 0.11). Interestingly, the association of severe malnutrition with all-cause mortality was maintained when age and tumor location were added as covariates in an adjusted multivariate Cox analysis (HR = 2.41, 95%CI = 1.03–5.59, *p* = 0.041).

## 4. Discussion

The GLIM criteria for malnutrition were recently developed to homogenize the diagnosis of malnutrition and include etiological variables, some of which are related to the presence and activity of the patient’s underlying disease. Multiple studies have investigated the diagnostic capacity of these new criteria for malnutrition [25,26,27]. Indeed, the GLIM criteria showed more sensitivity for the diagnosis of malnutrition than those previously described, such as the ESPEN [18]. However, data on their ability to predict the appearance of complications derived from malnutrition are scarcer in the literature, being in most cases retrospective studies or studies of hospitalized populations [16,19,21,22,28]. Furthermore, most of the work has been carried out in populations with a high incidence of malnutrition (head and neck or upper gastrointestinal) [15,20,22,27]. Low-risk populations for malnutrition, such as colorectal cancer patients, where the predictive ability may vary, have not yet been evaluated.

In our sample, no differences were observed in age or sex according to nutritional status. We did find, as expected, that patients with colorectal neoplasia had a lower prevalence of malnutrition than patients with tumors of the head and neck and upper digestive tract. Nutritional status was also not related to tumor stage, treatment time, or type of treatment. On the other hand, malnutrition was related to the stage of the disease, being more frequent in patients in progression.

Regarding the predictive capacity of these criteria during follow-up, the GLIM criteria were associated with a higher incidence of several events assessed. Patients with a diagnosis of malnutrition presented more frequently to the hospital emergency department, although statistical significance was not reached. However, significance was reached for the rate of hospital admissions, excluding elective admissions, whether for surgery, diagnostic procedures, or any other cause.

In terms of the risk of infection, no differences were observed for mild infection, but there were statistically significant differences in the rate of severe infection. These data are consistent with the literature. We hypothesize the existence of a multifactorial cascade involving different groups of lymphocytes, as well as immunoglobulins and cytokines, whose expression and/or development are increased by poor nutritional status [29,30].

Malnourished patients had worse pain control than their non-malnourished counterparts. This is evidenced by the higher rate of poor pain control and the higher need for opioid treatment (or increased doses of opioids) in malnourished patients. Pain has been described in the literature as an etiological factor in malnutrition, especially in the case of abdominal or upper digestive tract pain [29,31,32]. However, there is little evidence that malnutrition favors the onset of pain or worsens pain control. The prospective nature of our study suggests that in our patients, nutritional status was the differential factor in terms of pain management and not the other way around. This may be explained at least partially because malnutrition and consequent sarcopenia lead to a greater perception of pain in patients, resulting in poorer pain control [30,31].

The GLIM criteria were able to predict the occurrence of toxicities related to cancer treatment, with results consistent with the literature available to date [33,34,35]. These toxicities have a major impact on patient outcomes, leading to emergency department visits and admissions, as well as the need to discontinue treatments (temporarily or permanently), depending on the severity of the toxicity. This need to lower or suspend treatments was also observed in our malnourished patients. Toxicities and consequent dose reductions and interruptions result in suboptimal therapy, which can have a direct impact on patient survival.

Patients diagnosed with malnutrition by the GLIM criteria had a higher rate of tumor progression, as assessed by the RECIST criteria. Again, although both situations may be the cause and consequence of each other, the prospective design of the study suggests that it was malnutrition that caused an increased risk of progression.

Finally, and probably because of all the factors described above, the diagnosis of malnutrition by the GLIM criteria also predicted earlier and higher mortality. According to multivariate analysis, the ability of the GLIM criteria to predict mortality was independent of age and tumor location. Given the different populations studied, these results suggest that the GLIM criteria maintained their predictive ability for different prevalences of malnutrition. This confirms the usefulness of the GLIM criteria in populations with different levels of malnutrition risk. The relationship between malnutrition and reduced survival is established, and our work confirms that the GLIM criteria maintained this predictive ability.

Taking all these data into account, our study is one of the first in the literature to confirm the ability of the GLIM malnutrition criteria to predict the evolution of cancer outpatients, both during the tumor disease and the complications derived from it. Our work, in addition to confirming the practical utility of the GLIM criteria, reinforces the importance of the early diagnosis and treatment of malnutrition, given its impact on the outcomes of cancer patients.

The main strength of our study is its prospective design. In addition, the presence of different tumor sites increases its applicability. The variability in the prevalence of malnutrition among the different populations also increases the external validity of our results. The main limitations are the single-center nature of the study and the reduced sample size, which limits the possibility of performing subgroup analyses. Pain tolerance is a subjective variable, so its interpretation may be influenced by bias. The relatively short follow-up period may also be a limitation of the study.

## 5. Conclusions

The GLIM criteria for malnutrition predicted overall survival in cancer outpatients as well as the rate of complications, such as severe infections, admissions, and pain. In addition, patients diagnosed as malnourished by the GLIM criteria had worse tolerance to cancer treatments with higher toxicities, requiring a chemotherapy dose reduction and even treatment discontinuation. This predictive capacity makes the GLIM criteria a very useful tool in clinical practice, making it possible to identify patients at high risk of poor outcomes.

## Figures and Tables

**Figure 1 biomedicines-10-02201-f001:**
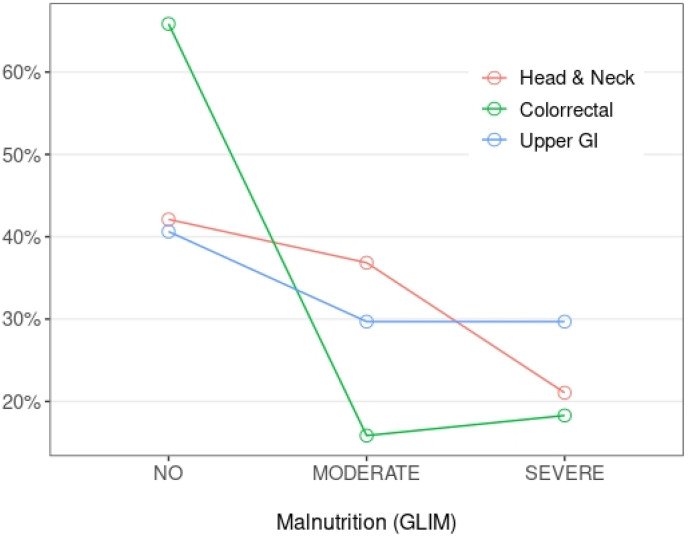
Malnutrition prevalence according to GLIM criteria and tumor location.

**Figure 2 biomedicines-10-02201-f002:**
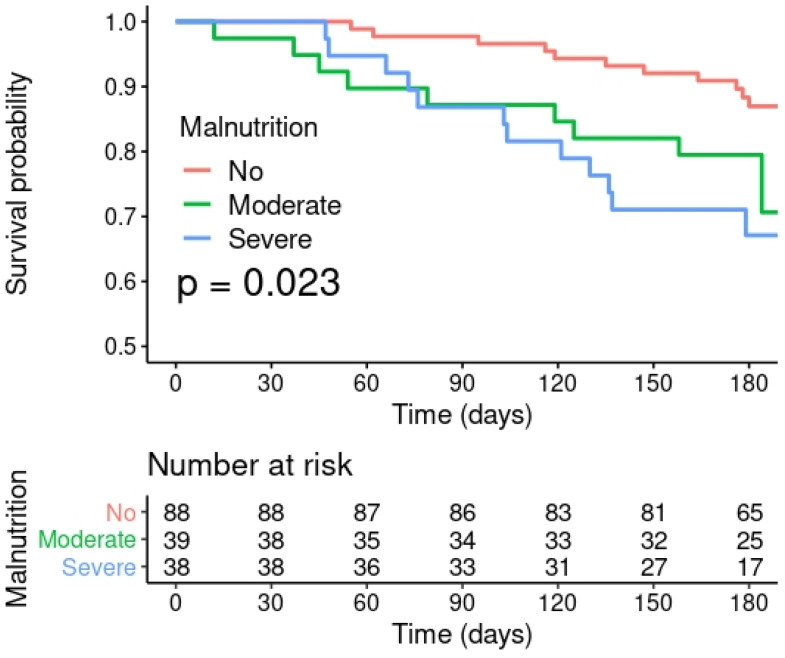
Kaplan–Meier survival curves grouped by nutritional status according to GLIM criteria.

**Table 1 biomedicines-10-02201-t001:** Demographic and tumor-related clinical variables.

Variables	Patients *N* = 165
**Sex *N* (%)**	
Male	107 (64.8%)
Female	58 (35.2%)
**Age (years)**	67.0 [60.0; 74.0]
**Localization *N* (%)**	
Head and neck	19 (11.5%)
Colorectal	82 (49.7%)
Upper GI	64 (38.8%)
**Stage *N* (%)**	
Localized	51 (30.9%)
Metastatic	114 (69.1%)
**Treatment *N* (%)**	
Chemotherapy	137 (83.0%)
Chemotherapy + RT	20 (12.1%)
Immunotherapy	4 (2.42%)
Targeted therapy	4 (2.42%)
**Duration of treatment (months)**	3.00 [1.00; 7.00]
**Current state *N* (%)**	
Response	51 (30.9%)
Stable	68 (41.2%)
Progressing	46 (27.9%)

Data are shown as median [IQR] for continuous variables and number of cases (%) for categorical variables.

**Table 2 biomedicines-10-02201-t002:** Prevalence of phenotypic and etiologic criteria according to the GLIM consensus in oncology patients with different tumor locations.

GLIM Criteria	Head and Neck	Colorectal	Upper GI	*p*
	*N* = 19	*N* = 82	*N* = 64	
**Phenotypic criteria**				
Non-volitional weight loss				0.018
No	13 (68.4%)	53 (64.6%)	25 (39.1%)	
Moderate	4 (21.1%)	15 (18.3%)	18 (28.1%)	
Severe	2 (10.5%)	14 (17.1%)	21 (32.8%)	
Low BMI				0.179
No	15 (78.9%)	77 (93.9%)	56 (87.5%)	
Moderate	2 (10.5%)	3 (3.66%)	3 (4.69%)	
Severe	2 (10.5%)	2 (2.44%)	5 (7.81%)	
**Reduced muscle mass**				0.538
No	11 (57.9%)	62 (75.6%)	44 (68.8%)	
Moderate	5 (26.3%)	12 (14.6%)	11 (17.2%)	
Severe	3 (15.8%)	8 (9.76%)	9 (14.1%)	
**Etiologic criteria**				
Reduced food intake or assimilation	11 (57.9%)	34 (41.5%)	34 (53.1%)	0.244
Inflammatory condition	2 (10.5)	21 (25.6)	23 (35.9)	0.077
GI chronic condition	5 (26.3%)	3 (3.66%)	20 (31.2%)	<0.001

Data are shown as number of cases (%). *p*: *p*-value for the difference.

**Table 3 biomedicines-10-02201-t003:** Impact of malnutrition by GLIM criteria on follow-up.

	No Malnutrition	Malnutrition	OR	*p*
	*N* = 95	*N* = 70		
Emergency Room admission	42 (44.2%)	41 (58.6%)	1.78 [0.95; 3.35]	0.071
Hospitalization	19 (20.0%)	35 (50.0%)	3.95 [2.00; 8.01]	<0.001
Severe infection	12 (12.6%)	23 (32.9%)	3.34 [1.54; 7.57]	0.002
Mild infection	26 (27.4%)	23 (32.9%)	1.30 [0.66; 2.55]	0.452
Poor pain control	26 (27.4%)	40 (57.1%)	3.50 [1.83; 6.83]	<0.001
Increase in opioid dosage	19 (20.0%)	37 (52.9%)	4.42 [2.24; 8.99]	<0.001
Toxicity	76 (80.0%)	64 (91.4%)	2.61 [1.03; 7.65]	0.044
Gastrointestinal toxicity	49 (51.6%)	44 (62.9%)	1.58 [0.84; 3.00]	0.153
Hematological toxicity	19 (20.0%)	25 (35.7%)	2.21 [1.10; 4.52]	0.027
Neurological toxicity	37 (38.9%)	26 (37.1%)	0.93 [0.49; 1.76]	0.817
Decrease or discontinuation of treatment	60 (63.2%)	57 (81.4%)	2.53 [1.23; 5.44]	0.011
Tumor Progression	29 (30.5%)	32 (45.7%)	1.91 [1.00; 3.65]	0.049
6-month survival	83 (87.4%)	50 (71.4%)	0.37 [0.16; 0.81]	0.012

Data are shown as number of cases (%). OR: Odds Ratio; *p*: *p*-value for the difference.

## Data Availability

The data presented in this study are available on request to the corresponding author with the prior authorization of our Ethical Committee that can be obtained at https://www.iacs.es/investigacion/comite-de-etica-de-la-investigacion-de-aragon-ceica/ceica-evaluaciones-y-otras-presentaciones (accessed on 30 March 2022).

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
