# Peer review of "Diagnosis of Malnutrition According to GLIM Criteria Predicts Complications and 6-Month Survival in Cancer Outpatients"

_biomedicines, 2022, doi:10.3390/biomedicines10092201_

Round 1
Reviewer 1 Report
In my opinion, the manuscript was well prepared, so I suggest accepting it in actual form.
· The Introduction Section explains the design of the study. The Authors well justify the research topic.
· The study was carried out without methodological errors.
· The Descriptions of the results were correct.
· The presented figures and table were prepared precisely and also legible.
· The Discussion Section included the accurate reference of the results obtained to the studies of other authors.
· The Conclusions were well formulated.
Author Response
We appreciate the time and effort that the reviewer has dedicated to providing valuable feedback on our manuscript,
Reviewer 2 Report
The manuscript entitled “Diagnosis of malnutrition according to GLIM criteria predicts complications and 6-month survival in cancer outpatients” is very interesting. However, I think the manuscript must go through a major revision, in terms of writing and data presentation. At the present stage the manuscript does not meet the level of the journal. Authors must focus on the following points.
1. I suggest discussing the points of GLIM (Global Leadership Initiative on Malnutrition) criteria briefly in the introduction section of the manuscript.
2. I suggest not to break the paragraph so frequently with only two or three sentences in it. And bring the connectivity between the paragraphs. In short, authors must focus on the writing.
3. I am unable to correlate the data shown in figure 1 with its description mentioned in line 155-157. I could see that colorectal cancer patients have low malnutrition i.e., below 20 %. However, the authors say it's 37%. Similarly, I can’t see more than 50 % either in the case of head and neck tumors, or in those with upper GI cancer.
4. In figure 1, I recommend plotting standard deviation in order to measure variability within the group.
5. I also recommend providing the standard deviations in all the tables along with the numeric values in percentage.
6. All the tables need clear and more detailed legends to understand the data. For example, what is the value inside and outside the bracket?
Author Response
The manuscript entitled “Diagnosis of malnutrition according to GLIM criteria predicts complications and 6-month survival in cancer outpatients” is very interesting. However, I think the manuscript must go through a major revision, in terms of writing and data presentation. At the present stage the manuscript does not meet the level of the journal. Authors must focus on the following points.
- I suggest discussing the points of GLIM (Global Leadership Initiative on Malnutrition) criteria briefly in the introduction section of the manuscript.
Thank you for the indication. We have added the next paragraph in the introduction explaining briefly how the GLIM criteria are assessed, as you suggested.
“It is a two-step approach in which the risk of malnutrition is first evaluated and then, those patients at risk that meet at least one phenotypic criterion (weight loss, low body mass index, or low fat-free mass index) as well as one etiological criterion (decreased intake, malabsorption, or inflammatory condition) are diagnosed as malnourished.”
- I suggest not to break the paragraph so frequently with only two or three sentences in it. And bring the connectivity between the paragraphs. In short, authors must focus on the writing.
A thorough revision of the manuscript has been carried out in order to make all the necessary spelling, grammatical and stylistic corrections.
- I am unable to correlate the data shown in figure 1 with its description mentioned in line 155-157. I could see that colorectal cancer patients have low malnutrition i.e., below 20 %. However, the authors say it's 37%. Similarly, I can’t see more than 50 % either in the case of head and neck tumors, or in those with upper GI cancer.
The percentage expressed in the text corresponds to the overall percentage of malnutrition (which includes moderate malnutrition and severe malnutrition). We thank the reviewer for pointing us this ambiguity which is now corrected in the revised version of the manuscript to make it easier for the reader to understand.
- In figure 1, I recommend plotting standard deviation in order to measure variability within the group.
- I also recommend providing the standard deviations in all the tables along with the numeric values in percentage.
Percentages reflect categorical data and as categorical data, standard deviation is not applicable. However, to avoid miss-interpretation we have included the following paragraph in the description of the statistical analysis:
“Categorical data were summarized by showing the number of individuals who fell into each category (e.g. malnutrtion) and creating a relative frequency. The relative frequency of a given category is the frequency (number of individuals in that category) divided by the total sample size, multiplied by 100 to get the percentage”
- All the tables need clear and more detailed legends to understand the data. For example, what is the value inside and outside the bracket?
We appreciate the reviewer’s observation. In some of the tables the legend explaining the interpretation of the data was missing. All of them have been revised and the legend has been added to the missing ones. Thank you very much for your careful observation.
Reviewer 3 Report
The authors have tried to use the GLIM criteria and performed a prospective study instead of doing a retrospective study as has been done previously. Using this study they tried to show that such a criteria of malnutrition could help to predict overall survival and extent of complications in people. The authors did find some correlations to identify high risk patients in clinics.
Questions to the authors:
1. In the text the authors say 64% are female but in the table they say it is males. Which one is correct as that can have a big impact on the interpretation?
2. The authors do make some conclusions but the different effects that the authors look at can be affected by multiple things besides just malnutrition status. The authors must acknowledge in the text that the sample size is too small to make such conclusions since there could be multiple complications in these people that would affect their survival and levels of complications.
3. Pain tolerance is a very subjective criteria and again should be acknowledged in the text.
4. My concern is that the different criteria used to select the patients are very diversified and at that age and stage of disease there are multiple complications happening. It is tricky to conclude that such a criteria could help predict response in clinic.
5. For inflammation criteria, did they do any serum cytokines measurements?
5. There are spelling errors in text. Please do a spell check on tables also.
Author Response
The authors have tried to use the GLIM criteria and performed a prospective study instead of doing a retrospective study as has been done previously. Using this study they tried to show that such a criteria of malnutrition could help to predict overall survival and extent of complications in people. The authors did find some correlations to identify high risk patients in clinics.
Questions to the authors:
- In the text the authors say 64% are female but in the table they say it is males. Which one is correct as that can have a big impact on the interpretation?
Thank you for notifying us of the error. As you say, it has a big impact on the interpretation of the data. It has now been corrected, and the correct figure is that 64% of patients were male.
- The authors do make some conclusions but the different effects that the authors look at can be affected by multiple things besides just malnutrition status. The authors must acknowledge in the text that the sample size is too small to make such conclusions since there could be multiple complications in these people that would affect their survival and levels of complications.
Thank you very much for your comment. The authors agree that the sample size is relatively small. We have emphasized this point in the limitations of the study. However, given that these are positive results, it is less likely that the result is influenced by the sample size. In addition to reaching statistical significance, the effect size is relevant and the results are consistent. This strengthens the conclusions in our view, and although they should be taken with due caution, we believe that the results are robust. This, together with the prospective design, leads us to believe that the differences observed may be largely due to the nutritional status of the patients.
- Pain tolerance is a very subjective criteria and again should be acknowledged in the text.
We fully agree with this assessment and have included this fact in the limitations section of the study. However, given that there is no objective method for assessing pain, we believe that this does not affect the quality of the study, as there is no gold standard for assessing this point with greater methodological quality.
- My concern is that the different criteria used to select the patients are very diversified and at that age and stage of disease there are multiple complications happening. It is tricky to conclude that such a criteria could help predict response in clinic.
The reviewer is right. Our study is a real clinical practice study, so it is impossible to control for all confounding variables. We have tried to prioritize external validity over internal validity in our work. Therefore, we selected different tumor sites based on their nutritional risk. We have done so because most of the studies carried out are in very specific populations. In these cases, it is difficult to extrapolate the results to clinical practice. As for age and tumor location, the survival analysis was performed by adjusting the model for these two variables.
- For inflammation criteria, did they do any serum cytokines measurements?
Cytokine values have not been used for the assessment of inflammation. This would certainly provide valuable information for the assessment of the inflammatory situation. However, those parameters are not used routinely, so they would not be useful common clinical practice.
- There are spelling errors in text. Please do a spell check on tables also.
A thorough revision of the manuscript has been carried out in order to make all the necessary spelling, grammatical and stylistic corrections.
Round 2
Reviewer 2 Report
I think the current manuscript is suitable for publication because authors have modified the manuscript keeping the reviewers point in mind. The reply to the comments are satisfactory to me. Hence, as per my opinion it can be accepted in the current format.
Reviewer 3 Report
The authors have addressed my concerns well and have tried to emphasize the limitations of such a dataset and the complications. I am glad that the authors are aware of the limitations and have mentioned them in the text.